# Nudging Consumer Behavior with Social Marketing in Portugal: Can Perception Have an Influence over Trying Insect-Based Food?

**DOI:** 10.3390/insects14060547

**Published:** 2023-06-12

**Authors:** Rocsana Bucea-Manea-Țoniș, Oliva M. D. Martins, Laura Urdeș, Ana Sofia Coelho, Violeta-Elena Simion

**Affiliations:** 1Doctoral School, National University of Physical Education and Sport, 060057 Bucharest, Romania; rocsense39@yahoo.com; 2Instituto Politécnico de Bragança (IPB), Campus de Santa Apolónia, 5300-253 Bragança, Portugal; oliva.martins@ipb.pt (O.M.D.M.); anasofia.coelho@ipb.pt (A.S.C.); 3Faculty of Veterinary Medicine, Spiru Haret University, 030352 Bucharest, Romania; ushmv_simion.violeta@spiruharet.ro; 4GOVCOPP-UA, University of Aveiro, 3810-193 Aveiro, Portugal

**Keywords:** social marketing, behavior, insect-based food, neophobia, perception

## Abstract

**Simple Summary:**

The world population has been continuously growing, along with life expectancy. More people living longer results in greater demand for different resources, such as water and energy, but also for food. It becomes imperative to ensure basic needs, including food of good quality. In this sense, providing food for all increases the demand for human and animal protein. Considering that food production, namely animal protein, has been associated with negative impacts on the environment, it becomes crucial to identify more environmentally friendly alternatives to the growing demand for protein. Insect-based feeding is one of these alternatives. However, not all countries have accepted it. Neophobia and disgust are two major barriers. To overcome these barriers, we need to change behaviors. Social marketing campaigns can help.

**Abstract:**

Social marketing campaigns are widely used to inform, educate, communicate, and promote healthy behaviors that add benefits to the individual, but also to society and the environment. Considering the low cost and high quality of insect-based food, this research aims to identify the main factors which can be used by social marketing campaigns to help people to try new foods, such as insect-based food. Although it is considered an important alternative to protein, there are a few countries that have not experienced it. In many Western countries, insect-based food is perceived as being disgusting. Neophobia is also a barrier to trying these foods. The main goal is to analyze if social marketing campaigns might influence perception (familiarity, preparation, visual, and information). Our model proves this assumption because we obtained high path coefficients, indicating that perception influences social beliefs, individual beliefs, and consumption intention. Thus, they will increase the consumption intention.

## 1. Introduction

The world population has grown and will reach 9.7 billion in 2050 [1]. It is crucial to ensure that basic needs are met, including the provision of good quality protein for human and animal food. In response to the estimated population growth, many researchers have already considered edible insect-based food as one of the alternatives for food and feed, a fact that can be supported with solid arguments. However, if this scenario does not exist, are edible insects a viable source of food? The authors consider that the current option of using insects as food does not only serve this purpose. It can be sustained in conditions of catastrophic climate change, war–associated events, and other events associated with problems related to the size of the human population or with the trend in the natural feeding of animals (e.g., the feeding of birds on the ground), which involve the natural consumption of insects [2]. Finally, from a food safety point of view, research is only at the beginning regarding contamination (microbial, chemical, parasitic) of these food sources [3], as well as the allergic reactions caused by these foods to some consumers. Nevertheless, the nutritional value of insects is high.

Insects have been studied as an alternative protein supply. Depending on the type and development stage, insects have significant protein content [4]. Insects have a protein content ranging from 10% to 25% (fresh weight) or 35% to 60% (dry weight) [5]. However, recent research has shown that processing significantly reduces the protein content of palatable insects [6]. Hopefully, insect-based food production will be affordable for most people, and the quality should be high, which means that this would be an alternative to human and animal food. However, there are countries where people are reluctant to implement insect consumption. Changing eating behavior on its own is a difficult task. Acceptability of these foods is even more difficult in Western European countries where people do not have this habit, because of barriers such as disgust and neophobia.

Social marketing campaigns can use the same principles, techniques, and tools of marketing [7] to inform, educate, and communicate value [8], as well as to promote good behavior [9]. In addition, social marketing campaigns can provide a useful “toolkit” to change behaviors [10], with benefits to the individual as well as to society [11]. Nevertheless, changing food behavior is complex [12]. Based on these principles, in relation to changing behavior regarding the decision of trying insect-based products as a new food, the main question is: can social marketing campaigns help people to try insect-based food as a new alternative to conventional food? In order to answer this question, the objective of this research is to identify the main factors which can be used by social marketing campaigns to help people to try new foods such as insect-based food.

Attitude and intention are two important constructs by which to analyze individual behavior [13], as well as perception [14]. Perception can be used in social marketing campaigns in a variety of ways, including: (1) initiatives that carefully frame the problem or behavior they are trying to change in order to impact how the target audience perceives it; (2) frequently makes use of the ability of social norms to shape views; (3) uses persuading messaging and communication techniques to shape perceptions; (4) features positive testimonials, success stories, or endorsements from powerful people or organizations to influence perception.

According to the Theory of Reasoned Action (TRA), intention precedes behavior, and individual beliefs and social beliefs influence intention [15]. This research considered the assumptions of TRA, as well as the relevance of perception regarding insect-based food [13,14]. The structure of the work presents the framework of factors of influence regarding insect-based food. This framework then helps to develop an instrument and the methodology used in order to obtain the study results. The discussion and conclusion are presented at the end.

## 2. Insects Used as Food around the World

There is no doubt regarding the importance of some considerable environmental benefits resulting from insect rearing for food. Insects have high feed conversion efficiency, less water and land are utilized in this activity, and fewer greenhouse gas emissions are produced [16]. It is foreseen that edible insects, as well as other “novel foods”, will contribute to the objectives of the Green Deal and Farm to Fork strategy [17].

The EC Regulation No 258/97 from 27 January 1997 concerning novel foods and novel food ingredients strikes a balance between food innovation and food safety, creating the opportunity for innovative foods to be launched to the EU consumer and guaranteeing their safety. The regulation concerns food items that were not significantly consumed in the EU before 15 May 1997, including insects and algae, underpinning the principles that they must be safe for consumers, properly labeled, and of equal or (better) nutrition quality compared to conventional food [18]. Despite its huge importance related to alternative proteins, there are still people in many Western countries who consider it disgusting. Neophobia is an additional barrier to trying these foods. Nevertheless, the quality of insect-based food is a motivating factor to try it [19].

Insects can supplement traditional food items such as fishmeal, soy, maize, and grains [20]. Amongst the insects considered with the greatest potential for large-scale pro-duction are the larvae of the black soldier fly (*Hermetia illucens*), the common housefly (*Musca domestica*), and the yellow mealworm (*Tenebrio molitor*) [21,22]. Since January 2023, insects have been used in powder, partially defatted, added to previously tested products, frozen, and dried. Other insect species are also being investigated for food purposes. In January 2023, the EU Commission authorized the placing of the lesser mealworm (*Alphitobius diaperionus*) on the food market. Eleven applications were considered valid and a safety evaluation by EFSA is currently being performed for each of them. It can be found in stored food grain products, such as flour. House crickets (*Acheta domesticus*) are also intended to be marketed as a food ingredient in several food items, allowing for the placement of the species on the EU market under certain conditions of use [23].

The quality characteristics of insect-based food should be a motivating factor to try it. In terms of nutritional value, the composition of fatty acids in insects is comparable to that of fish and chicken [24]. According to Toti [24], there are possible concerns associated with insect ingestion, such as the possibility of the bioaccumulation of methylmercury (MeHg) in dragonflies, although the amounts of oxalate, phytic acid, phenol, and tannins found in edible insect species are below the toxicity limits for human consumption.

### 2.1. Behavior Influence over the Consumption of Insect-Based Food

In Thailand, it is common to eat insects. Because of their great nutritional value and affordable manufacture, entomophagy is considered an alternative source of protein; but, in many civilizations, consumer acceptability of insect-based cuisine remains a significant obstacle. People’s propensity to consume insect-based dishes is influenced by social beliefs. Thus, these foods are rather consumed in social contexts (cafes, bars, pubs, food festivals), both alone and with social partners (friend, family, acquaintance, lover) [25,26].

Entomophagy is influenced by perception factors such as visual aspects, preparation, and information (using names rather than images), and association with positive arousing emotions (joy, excitement, romance, tranquility). The adoption of innovative insect-based goods might be increased by including messaging regarding the advantages of eating insects during testing [27]. Consumer acceptability of insect-based foods may be increased by their perception [28].

The attitude, subjective norm, and perceived behavioral control of the subjects all significantly contributed (80%) to their desire to eat insect-based meals. Even with the goal to consume food derived from insects, culinary neophobia appeared to be adversely associated [27,29,30]. According to the theory of planned behavior (TPB), attitudes (A) are characterized as both favorable and unfavorable assessments of behavioral intention. According to Ajzen (2005) [31] and Quine et al. (2000) [32], attitudes are driven by a person’s views, an individual’s beliefs regarding the effects of their own conduct, and how strongly they hold those ideas and beliefs. The term “subjective norm” (SN) refers to how social conditioning affects the desired behavior. Individuals frequently assume that those close to them approve or disapprove of their planned behavior. People are under more social conditioning to alter their behavior if they think that those around them consider their intended behavior suitable. 

The concept of perceived behavioral control (PBC) refers to people’s perceptions (individual beliefs) of their capacity to carry out the desired behavior founded on past encounters or hearsay information from sources including friends, family, coworkers, and social networks, which is often greater the more people have access to resources and opportunities and the less they see barriers to changing their behavior [31]. Social marketing professionals engage in a deliberate strategic procedure designed to introduce one small community to innovative cuisine beforehand if its readiness to ingest insect-based meals relies on embracing mnemonic shortcuts tied to cultural preconceptions. This ought to be carried out prior to spreading the novel food cuisine using educational programs to increase consumer awareness of consuming insect-based foods [33].

The future growth of entomophagy, as predicted by the respondents, as well as their intentions to try insects as food, the willingness to repeat an experience, the frequency of intake, and the quantities ingested are real and have been revealed by different studies published so far. The near-universal knowledge (96%) of entomophagy, widespread prior experience of eating insects (67%), and near-universal desire to eat insects again among those who have already done so (90%), all indicate that Western Europe may be a region where entomophagy has a very promising future. This is especially true for young men who care about the environment; however, product adoption presents difficulties due to issues with appearance, culture, and texture, requiring customized marketing strategies [34]. Penedo et al. [35] stated in a recent study that most participants in Switzerland indicated curiosity as the most likely justification for eating insect-based meals. Disgust was the most frequent pretext for avoiding such articles [35]. 

### 2.2. Insect-Based Food and Respective Factors of Influence

Consumer perception [36] can be influenced by sensory liking [37], which is also a very important variable to understand the decision of choosing insect-based food [14]. Familiarity [38] is another important factor. A consumer is more inclined to taste familiar food. It can be understood as information regarding the risks versus benefits [39]. Familiarity of the food represents a preference for a type of product [40], and it refers to foods that an individual usually eats [41].

Food color and food presentation [42] are important factors influencing willingness to try insect-based food [13,14]. This research considers that the visual appearance of insects does not encourage consumers to try them [38]. Insect-based food prepared as salty food [42] is considered more acceptable [38,43,44]. According to these authors, consumers consider the testing of sweet insect food less acceptable or even inappropriate [38,45,46]. In addition, the kind of food (e.g., traditional, authentic, ethnic, exotic, or artisanal) has an influence on the decision of trying insect-based food [47]. In this sense, being familiar with the food prepared according to one’s preferences, and especially when insects are not readily visible in the food, were considered important factors of perception that should be accounted for due to the increase in the predisposition to eating insect-based food.

From a different perspective, Martins’ research found that consumers preferred cooked, cold, and sweet insect-based foods [13]. The methods and ingredients were also considered important [13,14]. Thus, this research also considers it important to understand the influence of preparation on this decision. In this investigation, the preparation of insect-based food includes whether the food is cooked, salty or sweet [42], cold or hot, as well as the methods and respective ingredients.

Social pressure, as well as recommendations from friends, colleagues or family, can influence the intention to try insect-based foods [13]. However, regarding the insects themselves, one of the most relevant barriers is people being disgusted and neophobia, leading to negative effects on the willingness to try insect-based food [38]. 

The concept involves the fear of trying insect-based food. Individuals avoid eating insect-based food, and neophobia influences their eating behavior [48]. According to the authors, neophobia has a negative impact on the consumption of good foods, and people are not inclined to try it. Thus, beliefs and attitudes also have a negative association [13,38]. The same applies to the lack of information [38] and repugnance [42]. On the other hand, the information on the positive impact of eating insects, such as sustainability or the benefits for the environment, influences the individual’s predisposition to eating insect-based food [38]. The environmental impact increases the probability of choosing insect-based food. This research assumes that information helps to bypass these negative associations. Thus, familiarity with the food, the preparation of food, visual aspects, and information are important variables that influence sensory perception, which influence individual beliefs, social beliefs, and intention. Furthermore, individual beliefs and social beliefs influence intention. All variables are considered important to understand the decision of trying insect-based food. Table 1 presents the concept and respective indicators that influence the decision of trying insect-based food.

Based on these variables, indicators, and the respective relationship between them, this research proposes a model by which to evaluate this relation of influence. Figure 1 shows the model that represents these influences.

These variables and respective indicators, as well as the relation of influence, were analyzed through this behavioral model. The objective was to identify the main factors which could be used by social marketing campaigns to help people to try new foods, such as insect-based food.

Aimed at educating people regarding the long-term advantages, information about entomophagy either increases interest in eating insects or increases favorable feelings during testing, as well as good emotions. In order to promote insects as a food source, it may be effective to find customers who have a keen interest in insect-based products, which might be fundamental in social marketing campaigns [52]. 

## 3. Materials and Methods

Regarding the acceptance of insect-based foods in countries where people would be reluctant, attempts to change behavior would be a challenging task. Social marketing campaigns could prove a helpful tool that may assist in changing consumer perception towards new foods.

This research was grounded on the TRA; fundamentals considering that individual and social beliefs can influence intention [13], but also the relevance of perception [14]. In this investigation, perception is evaluated through visual aspects [13,14,38], information regarding the consequences (or benefits), as well as familiarity [39,40,41], and preparation [14]. The last assumption is that perception can influence the individual, their social beliefs, and intentions. These assumptions were also based on previous investigations.

### 3.1. Research Aim and Approaches

In our effort to identify the main factors which can be used by social marketing campaigns to help people to try new foods, such as insect-based foods, a research methodology was set up. A questionnaire was the instrument of the research. The first part of the questionnaire was focused on understanding the social demographic data regarding the sample. The second part of the questionnaire was thought to measure the variables and respective influences. In this sense, to evaluate the significance of the factors, an instrument was developed to allow for the implementation of quantitative research. 

### 3.2. Developing the Instrument

The intention was defined as a choice with commitment [50]. While social beliefs were conceptualized as social pressure, individual beliefs were cognitively defined, favoring attitude, and consequently, intentions and behavior [51]. Finally, consumers’ perceptions [14,36] resulted from different factors such as visual aspects, information, familiarity, and preparation. 

This research created a survey on this topic with multiple-choice questions, and the responses were analyzed. Although these results are relevant, it does not allow for a generalization of the results. For questions requiring multiple answers, the Likert scale was used. To apply the questionnaire, and taking into consideration our previous work, higher education students were considered [53] at the Instituto Politécnico de Bragança (IPB) in Portugal. The chosen age was above eighteen years old. To assess the age of the sample, a question regarding a range of ages was added, to allow for the selection of the following age ranges. This way, controlling the age of the sample was possible; i.e., only students eighteen years old or older were able to fill in the questionnaire. It was based on experiences with other universities [13,14,53]. However, only one hundred and eleven records (answers from Higher Education Institutions, HEIs) were verified. 

As a result, this study is only preliminary research, and in order to undertake secondary research, a more in-depth investigation using a representative sample is required. The research utilized SmartPLS Software [54] version 3.0 (University of South Alabama, Mobile, AL, USA) to analyze the data, performing variable association and inferential analyses. To analyze the opportunities, difficulties, and utilities, a confirmatory factor analysis (CFA) approach was created.

The study evaluated the effects of each element, including the intention to consume insects, social beliefs, individual beliefs, perception of food, the visual aspect of the food, familiarity with a specific kind of food, preparation of food, and information acquired. By categorizing survey items into four variables and specifying the direction of their relationships, CFA enabled us to impose constraints on the model. The primary drawback of CFA is that it obscures the influence’s direction. Even so, it can quantify the effects of each element and subfactor, as well as demonstrate whether the model provides a good fit for the data and is consistent. The influence’s direction was also validated by a literature review, as can be observed in Figure 1, the result of multiple research studies on food themes. Each variable’s loading factors are determined by the model. This was the primary justification behind choosing the CFA over a predictive analysis, such as a regression model [55].

Based on several metrics that improve how well the model explains the variables and matches the proposed hypothesis, the software estimates the model saturation. In this regard, comparative indices such as Akaike’s information criterion (AIC) and the Bayesian information criterion (BIC) can be listed, as well as absolute indices such as the normed fit index (NFI) and relative indices such as the statistic value of chi-square (which allows inferential statistics) or the standardized root mean square residual (SRMR). Cronbach’s alpha was used to analyze the relevance of the specified latent components, while composite reliability, rho_A, and average variance extracted (AVE) were used to assess the model’s consistency. The variance inflation factor (VIF) criterion was used to evaluate the model’s multicollinearity.

These indicators of the questionnaire are presented in Table 2.

The questionnaire was tested before being applied in Portugal. It was available online. It was applied between October 2022 and March 2023. The survey was conducted online through Google Forms. The survey was distributed through Google Forms Survey. Considering the minimum age limit of eighteen years old, for ethical reasons, the sample was obtained through different Portuguese higher education schools. Justified by the fact that it was a questionnaire applied to humans, the ethics review and protocol application received ethics committee approval (No 135/15.05.2021). Our previous research empowered us to use a very similar survey, improved with different nuances. There were one hundred and eleven answers. 

The survey was evaluated by a panel of experts in psychology. They conducted two types of validations: (a) qualitative pretesting (they tested if the content was understood, and if it applied to persons who were not the target of the questionnaire, and (b) quantitative pretesting (they tested if any adjustments may occur for the persons who are the target of the questionnaire). Construct validity was based on reporting on already validated questionnaires, but a statistical approach was also applied: Pearson’s correlation coefficient for variables measured on a report/proportional scale, or Spearman’s correlation coefficient for variables measured on nominal or ordinal scales. An evaluation of the fidelity and internal consistency of the questionnaire was also conducted, to check if the items of the questionnaire contributed to the constitution of the significance of the questionnaire and if the questions of the survey “went together” and mirrored the same characteristics. A questionnaire is consistent when the items of which it is composed correlate, with the additive result of all the items. In this regard, the Cronbach-alpha coefficient and inter-item correlation matrix were calculated. In our case, the Cronbach-alpha coefficient was greater than the threshold of 0.70, meaning that our approach is correct. The data analysis used was the structure equation model using Smart PLS software [54,55].

## 4. Results

In order to determine the respondents’ socio-demographic characteristics, as well as their views and perceptions, this research designed and utilized a questionnaire. Among the most significant demographic information, it should be noted that (Figure 2):

The main results determine the following profile of the sample: female, high education level (high school or higher), and middle income. Regarding eating protein, the majority eat protein in every meal. The summary of these data is presented below:68% were males and 32% were females.39% were 46 to 55 years old and 24% were 18 to 25 years old.50% had the faculty or master completed, as well as 28% had attended high school and 20% had a PhD. Therefore, the respondents were well informed.37% earned €500,00 to €1000,00 per person per month, and 50% had salaries higher than 1000€.54% ate protein at breakfast, 96% ate protein at lunch, and 92% ate protein at dinner.In other words, 46% did not eat protein at breakfast, 47% ate animal protein or both, 68% ate both kinds of protein (animal and vegetable) at lunch, 27% ate animal protein for lunch, 67% ate both kinds of protein (animal and vegetable) at lunch, and 20% ate animal protein for dinner; most of them ate animal protein and vegetables, especially for lunch and dinner.Other significant data regarded their job: 71% were paid and were employed, 28% were entrepreneurs, and most of them had good salaries and a stable economic situation.

Given that there are many variables that might affect a person’s desire to try an unfamiliar cuisine, the primary goal of the study was to determine the traits and variables that affect people’s intentions to try new foods. The research used PLS-SEM to analyze the data. 

The PLS-SEM approach is nonparametric and makes no assertions regarding the minimal probability [55]. When the number of issues (questions) rises, the value of the Cronbach alpha index [56] often rises as well. This research highlights their main variables: perception of social beliefs, and individual beliefs that influence the intention of consuming insect-based foods. The research used PLS-SEM to design a path analysis. Path analysis is a statistical technique used to examine causal relationships among variables. It is a form of structural equation modeling (SEM) that focuses on the direct and indirect relationships between variables in a hypothesized model. Path analysis allows researchers to assess the strength and direction of relationships between variables and to test specific hypotheses regarding the relationships among variables.

Our data series generally exhibit a normal distribution in the descriptive statistics (Table 3), with small standard errors. There were a few exceptions in this study: age had a high standard deviation (1.45) because the population interviewed was rather old (39% were 46–55 years old). It was justified to involve all people in a common cause. The level of education (literacy or studies) had a rather high standard deviation (1.13) because only 2% graduated from primary school studies. Regarding eating protein at breakfast, there was a rather high standard deviation (1.23) because 46% of the people would not eat protein in the morning. Since the values for kurtosis and skewness were limited within the range [−1.96, 1.96], our data series had a normal distribution. Regarding eating protein at “Lunch”, this was an exception because it had high values for kurtosis (3.42), but these variations were negligible, so the authors proceeded to the interpretation under the assumption that the sample was representative.

The elements that make up the questionnaire were examined, developing questions to measure the variables with the objective of assessing their importance. The questions were designed by comparing each variant of responses with the sum of all responses (across the overall score). In this case, the survey was trustworthy and consistent.

There were two types of variables in our analysis: Four formative variables: *Familiarity*, *Preparation*, *Visual*, and *Information* that have influence on *Perception* and, consequently, on *Intention*.Two formative variables *Social Beliefs* and *Individual beliefs* that have influence on *Intention*.Two reflective variables: *Perception* and *Intention*.In addition, *Perception* has influence on *Individual beliefs*, *Social Beliefs*, and *Intention*.In our analysis, there were two types of variables: formative and reflective. Familiarity, Preparation, Visual, and Information have an influence on Perception, and respectively on Intention, and Social Beliefs and Individual Beliefs have an influence on Intention. The literature review states that Social Beliefs and as Individual Beliefs influence Intention, which is the antecedent of the consumption. Figure 3 presents the PLS-SEM analysis using SmartPLS software version 3.3.9.

As a result, among the goals of the component analysis was to discover and remove these items or to alter them in accordance with the assessed characteristic, keeping items that made no or had little impact on the final score or that instead showed an opposite direction from this one would have been pointless. In order to create a model that was representational of this, some variables (Perc_prepared, Visual_color, Inf_society, Prep_cold, Prep_prepared) were eliminated, because they had very small values and a meaningless impact on the model.

The path coefficients of our model showed that insect-based food perception influences Social Beliefs (0.624), Individuals beliefs (0.887), and Intention of consumption (0.627). The model from the literature review states that *Individual* and *Social Beliefs* influence *Intention*. Our model found that *Perception* influences *Individual* and *Social Beliefs*, as well as *Intention*, and *Individual* and *Social Beliefs* also influence *Intention*, which means they have indirect effects.

The path coefficients prove that perception is influenced by *Familiarity (0.194), Preparation (0.227), Visual (0.283)*, and *Information (0.321). Familiarity* is mainly influenced by *Fam_daily with a very high loading factor* (*LF* = *0.841*), followed by *Fam_experiment* (*LF* = *0.128*) and *Fam_condiments* (*LF* = *0.080*). *Preparation* is mainly influenced by *Prep_salty* (*LF* = *0.451*) and *Prep_ingredients* (*LF* = *0.400*), and second by *Prep_cooked* (*LF* = *0.244*) and *Prep_processed* (*LF* = *0.128*). The Visual perception is mainly influenced by *Visual_presentation* with a very high *LF* = *0.879* and by *Visual_see* (*0.157*). The *Information* that influences *Perception* mainly depends on *Inf_health with a high LF* = *0.771* and *Inf_environment* (*0.254*).

*Social beliefs* are mainly *influenced* by *SB_friends&fam* (*LF* = *0.633*) and then by *SB_Try* (*LF* = *0.303*) and *SB_soc_acceptability* (*LF* = *0.196*).

*Individul beliefs* are mainly influenced by *IB_knowledge* (*LF* = *0.470*) and then by *IB_Preparation* (*LF* = *0.410*) and *IB_see* (*LF* = *0.197*).

*The Perception of food* is measured by *Perc_familiarity* (*LF* = *0.805*), *Perc_info* (*LF* = *0.852*), and *Perc_visual* (*LF* = *0.866*). Intention of consumption is measured by *Int_appearance* (*LF* = *0.914*), *Int_prepared* (*LF* = *0.862*), and *Int_familiarity* (*LF* = *0.731*).

It has a cyclical nature and operates by selecting things based on how they relate to the total score after evaluating the interconnections among the items, as well as among the items and the overall score in turn. The Cronbach’s alpha index value, which can vary from 0 to 1 [55], is the crucial requirement for this process. 

Most academics agree that a scale must have a value over 0.70 [57] in order to be regarded as consistent. Cronbach’s alpha, however, cannot be less than 0.60. Formative and reflecting estimation techniques may be distinguished using confirmatory tetrad analysis in PLS-SEM (CTA-PLS) [58]. Except for the bootstrapping strategy used to assess the predictive value of the model-implied tetrads [58], the study is similar to Bollen and Ting’s [59] confirmatory approach of assessing model-implied disappearing homologous pairs in the PLS-SEM context.

The development of the Partial Least Squares Algorithm (PLS) [60] should take place in three steps: (1) estimation of latent variable scores; (2) computation of outer weights and path coefficients; and (3) placement of variables [61] as a series of regression models regarding their weights [62]. SmartPLS 3.0 software may be used to create the Path Coefficient PLS Analysis (PLS) [55,63,64]. The data presented in Table 3 support the affirmations above as the indicators have higher values than the threshold.

Composite Reliability, R-Square, Rho A, Cronbach’s Alpha, and AVE were the criteria of the validation procedure for the latent variables. The research found that all of the variables matched all the validation requirements. The very high correlation (greater than 0.7) between most of the variables was the fundamental test conducted before designing the regression models. There were also medium correlations (between 0.5 and 0.69) between variables, with one exception, a small correlation between Preparation and Social beliefs (0.483) (see the correlation in Table 4).

Another issue was analyzing the indirect effects. Table 5 presents some small indirect effects between our variables.

A good match is explained by the Standardized Root Mean Square Residual (SRMR), which has a value smaller than 0.1 [65]. 

In order to calculate the discrepancy based on the Eigenvalue value, the terms d ULS and d G stand for the squared Euclidean distance and the geodesic distance, respectively. The Bentler and Bonett Index (NFI) or standardized fit index (NFI) [66]. After that, the NFI is calculated as one minus the Chi^2^. 

The NFI result is greater (i.e., better) with more model parameters [67]. The estimated values for SRMR, d ULS, d G, and chi-square are higher than the estimated values for saturated models. As a result, the authors can state that the model is reliable and supports these variables and their respective relationship, as presented in Table 6.

The correctness of our model is supported by all of the R Square values (Table 7). 

The R Square Adjusted prove that:●84.7% of the variance-dependent variable Individual beliefs is explained by the variance of independent variables (IB_knowledge, IB_Preparation and IB_see)●73.1% of the variance-dependent variable Intention is explained by the independent variable’s variances (Int_appearance, Int_prepared and Int_familiarity)●84.6% of the variance-dependent variable Perception is explained by the variance of independent variables’ variances (Perc_familiarity, Perc_info and Perc_visual)●38.5% of the variance-dependent variable social beliefs is explained by the independent variable’s variances (SB_friends&fam, SB_Try, and SB_soc_acceptability).

As a nonparametric strategy, PLS-SEM makes no assumptions regarding distributions. Nevertheless, the significance of outer weights, outer loadings, and route coefficients cannot be determined using parametric significance tests (such as those employed in regression analysis). To determine the significance of different outcomes, such as path coefficients, Cronbach’s alpha, HTMT, and R2 values, PLS-SEM instead uses a nonparametric bootstrap approach.

Sub-samples are randomly generated from the original data set in bootstrapping (with replacement). The PLS path model is then estimated using the subsample. Repeat this procedure until many random subsamples have been produced (e.g., 5.000). The standard errors for the PLS-SEM findings are calculated using the estimates from the bootstrapping subsamples. 

Bootstrapping is a re-sampling technique used to estimate the sampling distribution of one statistic. It involves repeated sampling from the original dataset with replacement to create multiple bootstrap samples. By analyzing these bootstrap samples, researchers can make inferences regarding the population from which the original dataset was drawn. Bootstrapping is particularly useful when the underlying distribution is unknown or when traditional assumptions about sampling distributions are not met.

Every construct’s Variance Inflation Factor (VIF) was used to evaluate the relevance of each variable. VIF is a measure of multicollinearity in regression analysis. It quantifies the extent to which the variance of the estimated regression coefficients is inflated due to high correlations among the predictor variables. VIF values greater than 1 indicate the presence of multicollinearity, with higher values indicating more severe multicollinearity. High VIF values can lead to unreliable and unstable regression coefficient estimates. There is no evidence of collinearity between the variables because the VIF is lower than the recognized maxima limit of (5). To assess the importance of the variables, the Variance Inflation Factor (VIF) of each construct was run using 1000 samples. With the aid of SmartPLs software, a bootstrapping reliability of 95% was accomplished [63]. Table 8 provides a summary of the findings.

SmartPLs software computes standard errors, t-values, and confidence intervals to determine the relevance of PLS-SEM data [63]. To determine the significance of the PLS-SEM results, t-values, p-values, and confidence intervals are computed using this information [68]. T-values are greater than 1.96 and signify the coherence of the model [68]. The prerequisites listed before are met. Figure 4 shows the path coefficient analysis of the bootstrapping analysis.

In the original TRA model, the intention is formed by an individual’s social and individual beliefs. However, for this behavior, social beliefs do not result in a major impact, hence, it can be considered insignificant. If social beliefs were impactful, families and friends would be the main agents involved in the behavior change. Regarding individual beliefs, the research found that influence can be considered moderate. Knowledge and preparation influence individual beliefs. According to these results, the individual’s perception is considered to have the greatest influence on the intention to try new insect-based foods. The model shows the relevance of perception’s influence on individual and social beliefs, as well as on intention. 

By analyzing perception, it is possible to argue that information has the biggest impact on it. Information regarding health is the most important indicator of information. Should social marketing campaigns aim at increasing perception, it requires more information on health-related benefits to be provided. Visual factors have the second largest impact on perception. In this case, the visual aspects related to the presentation of food help individuals to try insect-based food. Third in terms of influence is preparation. Unlike the other variables, there are several influencing factors in the preparation. Ingredients and salted food seem to be more influential in the preparation, but the process itself and the fact that the food is cooked also influence it a little. Familiarity with the food has the lowest influence, but the difference is reduced. Daily use is the most important indicator of familiarity. Once individuals are habituated to eating certain types of food, the process of experimenting with new ingredients is facilitated. Considering the importance of visual aspects and familiarity with food, the research assumes that processed foods, such as insect-based flour, help in experimentation and, consequently, in changing habits.

## 5. Discussion

Social marketing is a strategic approach that utilizes marketing principles and techniques [7] to promote positive social change [9]. It aims at influencing people’s attitudes, behavior [15], and perceptions [13,14] for the benefit of society [11]. Social marketing campaigns are typically focused on areas such as public health, environmental conservation, education, and social issues. Changing food behavior is a complex task [12].

The key principles of social marketing include behavior changes and understanding the target audience. To achieve these objectives, social marketing campaigns employ a mix of marketing strategies that involve creating compelling messages, identifying appropriate channels for communication, and ensuring the accessibility and affordability of the desired behaviors. In addition, social marketing approaches involve research and respective evaluation.

This article sets the focus on understanding the target audience to promote behavioral change goals. The main pillar that might be used in social marketing can be nudging consumer behavior, a concept derived from behavioral economics that refers to subtly influencing people’s behavior without removing their freedom of choice. It involves persuasive techniques to guide individuals toward making choices. Nudging takes advantage of cognitive biases and heuristics that affect decision-making processes. In practice, nudging can be applied in various domains, such as encouraging sustainable behaviors (e.g., using energy-efficient appliances), promoting healthier lifestyles (e.g., encouraging exercise or healthy eating), and fostering financial wellbeing (e.g., promoting savings or responsible spending). 

While answering the survey, the respondents were challenged to realize and understand the phenomena of solving feeding problems by using some insect proteins. Exposure might be a first step in social marketing campaigns with the aim of nudging their behavior. 

In this case, perception plays a crucial role in consumers’ behavior and marketing. Consumers’ perceptions of products, brands, and marketing messages significantly influence their attitudes, preferences, and purchasing decisions. Marketers aim to shape consumers’ perceptions through various strategies such as branding, advertising, packaging, and product positioning. Additionally, social marketing campaigns often rely on influencing individuals’ perceptions to promote behavior change because they seek to shape attitudes, beliefs, and social norms surrounding specific issues to encourage desired behaviors and discourage negative attitudes. 

Social marketing campaigns utilize perception in different ways, such as:Framing: Social marketing campaigns carefully frame the issue or behavior they are addressing to shape how it is perceived by the target audience. By highlighting certain aspects or emphasizing specific consequences, they aim to create a perception that motivates behavior change [69].Social norms: Social marketing campaigns often leverage the power of social norms to influence perceptions. By highlighting the prevalence of certain behaviors within a social group or community, campaigns seek to create a perception of what is considered normal or desirable, thereby encouraging individuals to accordingly align their behaviors.Messaging and communication: Effective social marketing campaigns employ persuasive messaging and communication strategies to shape perceptions. By using compelling narratives, emotional appeals, and relatable storytelling, they aim to create a perception that resonates with the target audience and motivates them to take action.Social proof: Social marketing campaigns often showcase positive testimonials, success stories, or endorsements from influential individuals or groups to influence perception. By providing evidence of others who have adopted the desired behaviors and experienced positive outcomes, campaigns seek to establish social proof and encourage perception change.

Thus, social marketing approaches recognize the importance of perception in shaping behavior and utilize various strategies to influence individuals’ perceptions, attitudes, and beliefs to drive positive social change.

Ensuring food has always been a challenge, but it is also a necessity for every person. Although some people from the poorest regions are striving to live, or suffer from malnutrition [70], residents of more prosperous nations employ conventional sources of protein (fish, meat) or move to new sources of protein that have already become familiar (vegetable protein). Nonetheless, the continuous expansion of the world’s population, the total lack of drinking water and natural resources, crisis situations when the supply of food resources is difficult to achieve, for example, in pandemics, during war, etc., will inevitably lead to the fact that some species of insects will be used worldwide as supplementary food for humans. 

At the same time, insects are a traditional food [20,47] for many people, and recent studies have only shared these culinary experiences [20] with other nations that have not consumed them until now. Obviously, this process can be naturally carried out, or it can be set out with food safety conditions [18,23]. Based on the main factors that can be used by social marketing campaigns to help people to try new foods such as insect-based food, the main goal is to analyze whether social marketing campaigns might influence Perception [13,36] (Familiarity [38,39,40,41], Preparation [13,38], Visual [13,14,38], and Information [38]), as well the influence of perception on social norms [13], attitudes or individual beliefs [51], and intention [49,50]. This was the main goal of our research: to identify the levers through which social marketing campaigns can help people try to consume new foods in a geographical area where they are not a common food, in safe conditions [71] and with real nutritional benefits.

From a different perspective, several studies have highlighted the favorable effects of insect consumption on health. Edible insects may be beneficial to human health through their antioxidant, antihypertensive, anti-inflammatory, and immunomodulatory effects [72,73,74], and have antimicrobial properties [75], including against enteropathogenic *Salmonella*—the causative agent of enteritis, contributing to the normalization of intestinal microbiota [19].

Insect feeds are rich in fats, fiber, valuable macro- and microelements, and vitamins. These feeds contain a large amount of protein with a full amino acid composition [76,77,78]. However, for safety purposes, appropriate processing protocols need to be applied for the microbial load and anti-nutritive content to be decreased [19,79]. The human diet is developing, not only nutritionally, but ecologically friendly and healthy products are also in demand [80]. The visual characteristics of edible insects, information on their origin and safety, taste and texture, as well as familiarity [81], are all important factors in enhancing the consumption of insect-based food. 

In summary, the social context influences nutrition through multiple mechanisms. In other words, social factors can have a special influence on nutrition, such as an influence on the behavior of others that is easily adopted on the principle: “if others do it, I can probably do it, too” [82]. 

Berger and Wyss [83] have shown that individual beliefs regarding the social norm correlate with a willingness to eat an unprocessed insect, and even in the absence of concrete information regarding social norms, consumers construct norms based on other options (e.g., financial rewards perceived according to diet).

Insect-based food is considered an important alternative to protein. However, there are a few countries that have not experienced it. In many Western countries, there are people who consider it disgusting. Neophobia is a barrier to trying it. 

## 6. Conclusions

Aimed at identifying the main factors that can be used by social marketing approaches to help people try new foods such as insect-based food, this research found that *individual perception* has the greatest influence over the drive to try new insect-based foods. In fact, *perception* is influenced by *familiarity, preparation, visual factors*, and *information*, mainly regarding the benefits, which can be used by social marketing campaigns to nudge consumer behavior toward trying a (new) insect-based food.

Concerning perception, the results show that information has the greatest impact on the way an individual regards the consumption of novel food. Health information is the most important indicator of information. The second largest impact on perception is repented by visual factors, whilst training comes third in this study. 

This study also indicates that, on the one hand, *social beliefs* are mainly *influenced* by *friends and family.* On the other hand, *individual beliefs* are mainly influenced by *knowledge* and, secondly, by *preparation.* Finally, the *perception of food* is measured by *familiarity, information*, *visual aspects*, and *preparation*. Knowledge and training influence individual beliefs, and social beliefs indirectly influence one’s beliefs, which, in turn, impact the drive to consume novel foods, such as insect-based foods.

Our model shows that social beliefs influence individual beliefs, which, in turn, influence consumption intention. This is an indirect effect. However, for this behavior, social beliefs do not have a major impact, and it can be considered insignificant. Knowledge and training influence individual beliefs. According to these results, individual perception was found to have the greatest influence on the intention to try new insect-based foods. By analyzing perception, it can be argued that information has the greatest impact on perception. Health-related information is the most important indicator of information. Visual factors have the second largest impact on perception. Third in terms of influence is training.

The model proved this assumption because high path coefficients were obtained, which means that perception influences social beliefs, individual beliefs, and consumption intention. Thus, they are likely to increase the consumption intention. The authors conclude that descriptive social norms would only partially underlie the willingness of European Western countries’ inhabitants to consume insects.

The main limitations of this study are related to the objective, identifying the main factors that can be used by social marketing campaigns to help people to try new foods, such as insect-based food, as well as its theoretical foundation (assumptions based on TRA and previous studies). Furthermore, the methodology, data collection, and data analysis also had to be limited. Nevertheless, the investigation intended to involve as many people as possible, which means that the sample is not homogeneous, posing difficulties in the interpretation of the results.

## Figures and Tables

**Figure 1 insects-14-00547-f001:**
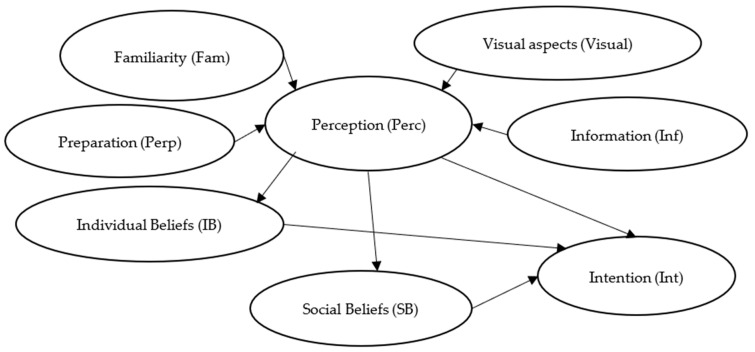
A proposal of a behavioral model by which to understand insect-based food as a new alternative for feeding people. Source: own elaboration.

**Figure 2 insects-14-00547-f002:**
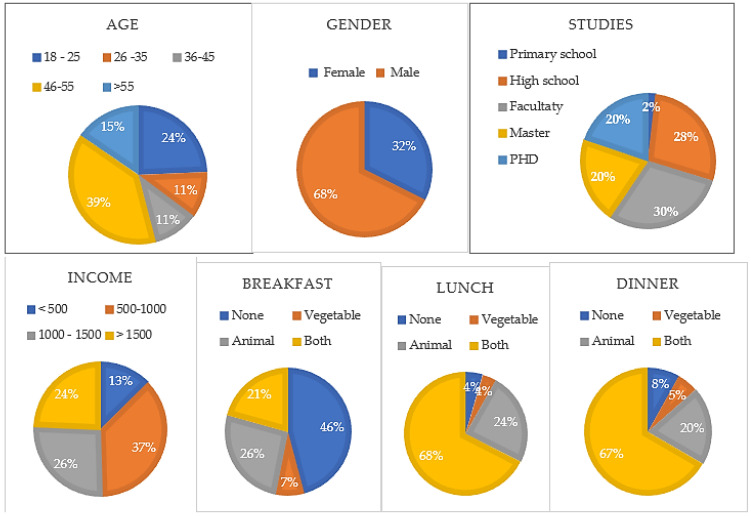
Analysis of social demographic data.

**Figure 3 insects-14-00547-f003:**
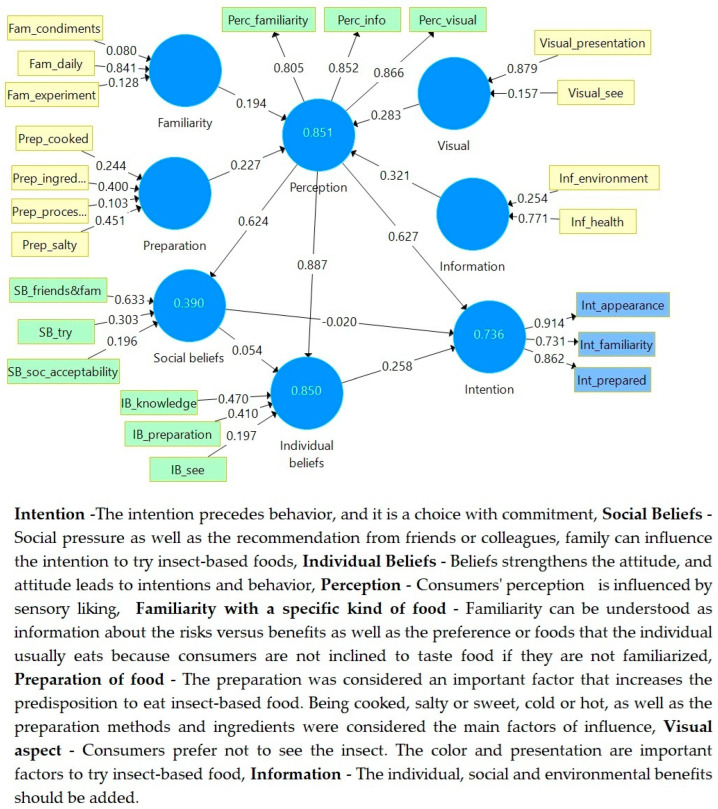
Path analysis of R2 using SmartPLS software. Source: SmartPLS analysis (reprinted from a free version of SmartPLS software, version 3.3.9, created on 31 March 2023) [54].

**Figure 4 insects-14-00547-f004:**
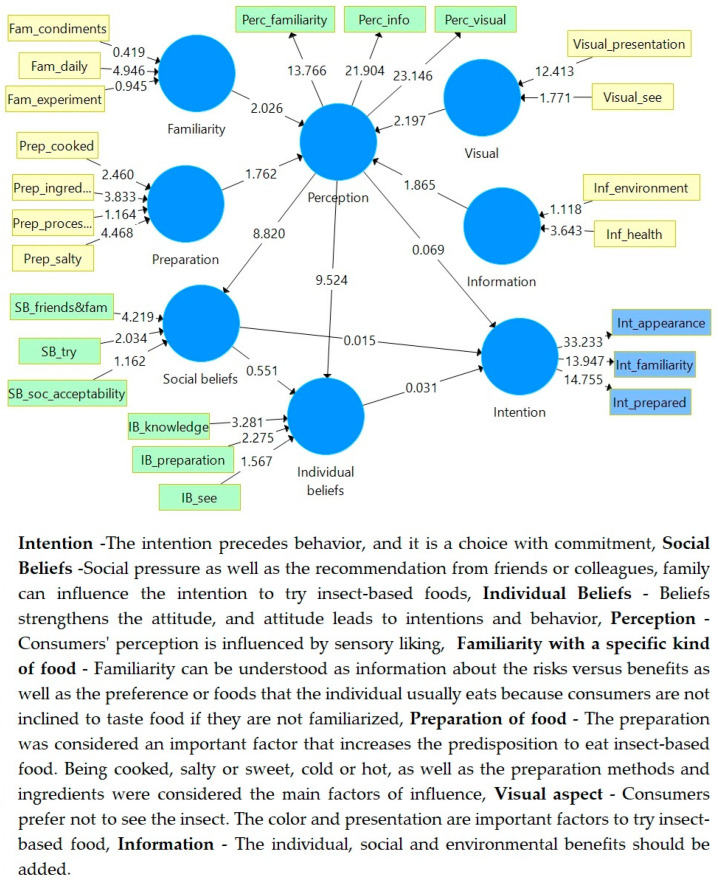
Path coefficient analysis of bootstrapping analysis (1000 samples). Source: SmartPLS analysis (reprinted from a free version of SmartPLS software, version 3.3.9, created on 31 March 2023) [54].

**Table 1 insects-14-00547-t001:** Variables of influence on the decision of trying insect-based food.

Variables and Indicators		Concept
Intention	Int	The intention preceedes behavior [49], and it is a choice with commitment [50].
Social beliefs	SB	Social pressure and recommendation received from friends, colleagues, or family can influence the intention to try insect-based foods [13].
Individual beliefs	IB	Beliefs strengthen the attitude, and attitude leads to intentions and behavior [51].
perception	Perc	Consumers’ perception [13,36] is influenced by sensory liking [13,37].
Familiarity with a specific food type of	Fam	Familiarity can be understood as information on the risks versus benefits [39] as well as the preference [40] for foods that the individual usually eats [41] because consumers are not inclined to taste food if they are not familiarized with it [38].
Preparation of food	Prep	The preparation was considered an important factor that increases the predisposition to eat insect-based food. Being cooked [13], salty [38] or sweet, cold or hot, as well as the preparation methods and ingredients [13] were considered the main factors of influence.
Visual aspect	Visual	Consumers prefer not to see the insect [13,14,38]. The color and presentation are important factors in trying insect-based food [13].
Information	Inf	The individual, social, and environmental benefits should be added [38].

**Table 2 insects-14-00547-t002:** Questionnaire applied.

Variables		Indicators
Intention to consume insects	Int	Q1.1 My intention to try insect-based foods depends on the information that I have Q1.2. My intention to try insect-based foods depends on their appearance Q1.3. My intention to try insect-based foods depends on how it is prepared Q1.4. My intention to try insect-based foods depends on familiarity with the food
Social beliefs on entomophagy	SB	Q7.1 I believe that recommendation from my family influences my intention to try insect-based food Q7.2 I believe that recommendation from my friends and colleagues influences my intention to try insect-based food Q7.3 I would try insect-based foods as long as other people have tried them first Q7.4 I believe that social acceptability influences my intention to try insect-based foods
Individual beliefs on entomophagy	IB	Q6.1 I believe that seeing the insect influences my intention toward experimentation of insect-based food Q6.2 I believe preparation influences my intention to try insect-based food Q6.3 I believe that having knowledge influences my intention to try insect-based food
Perception of food	Perc	Q5.1 My perception of insect-based foods is influenced by the visual aspects Q5.2 My perception of insect-based foods is influenced by the information I have Q5.3 My perception of insect-based foods is influenced by the way they are prepared Q5.4 My perception of insect-based foods is influenced by familiarity with the food
Visual aspect of food	Visual	Q3.1 I would try insect-based food as long as I don’t see the insect Q3.2 The color is an important factor to try insect-based food Q3.3 The presentation itself is an important factor to try insect-based food
Familiarity with a specific kind of food	Fam	Q8.1 The kind of food (e.g., traditional, authentic, ethnic, exotic, or artisanal) influences experimentation of insect-based foods Q8.2 Well-known condiments influence experimentation of insect-based foods P8.3 Preparing food in a similar way to daily food influences experimentation of insect-based products
Preparation of food	Prep	Q2.1 To try insect-based foods, I would prefer them cooked rather than dehydrated Q2.2 To try insect-based foods, I would prefer them to be served cold rather than hot Q2.3 To try insect-based foods, I would prefer them to be salty than sweet Q2.4 I would prefer to try insect-based foods if I knew how they were prepared Q2.5 I would prefer to try insect-based foods if I knew their ingredients Q2.6 I would prefer to try insect-based foods if they were processed (e.g., foods made with insect flour)
Information	Inf	Q4.1 I would be more inclined to try insect-based foods if I knew the benefits to my health Q4.2 I would be more inclined to try insect-based food if I know the benefits for the environment Q4.3 I would be more inclined to try insect-based food if I know the benefits for the whole society

Source: own elaboration.

**Table 3 insects-14-00547-t003:** Model validation criteria.

	Cronbach’s Alpha	rho_A	Composite Reliability	Average Variance Extracted (AVE)
Threshold	>0.7	>0.7	>0.7	>0.5
Familiarity		1.000		
Individual beliefs		1.000		
Information		1.000		
Intention	0.870	0.885	0.876	0.704
Perception	0.878	0.880	0.879	0.708
Preparation		1.000		
Social beliefs		1.000		
Visual		1.000		

Legend: Please see information in Table 2.

**Table 4 insects-14-00547-t004:** Correlation between variables.

	Familiarity	Individual Beliefs	Information	Intention	Perception	Preparation	Social Beliefs	Visual
Familiarity	1.000							
Individual beliefs	0.742	1.000						
Information	0.707	0.842	1.000					
Intention	0.743	0.823	0.832	1.000				
Perception	0.780	0.921	0.845	0.852	1.000			
Preparation	0.683	0.768	0.741	0.836	0.825	1.000		
Social beliefs	0.591	0.608	0.650	0.528	0.625	0.483	1.000	
Visual	0.719	0.801	0.769	0.893	0.852	0.804	0.521	1.000

Legend: Please see information in Table 2.

**Table 5 insects-14-00547-t005:** Indirect effects.

	Individual Beliefs	Intention	Social Beliefs
Familiarity	0.178	0.165	0.121
Individual beliefs			
Information	0.296	0.274	0.201
Intention			
Perception	0.033	0.234	
Preparation	0.209	0.194	0.142
Social beliefs		0.013	
Visual	0.261	0.241	0.177

Legend: Please see information in Table 2.

**Table 6 insects-14-00547-t006:** Model Fit.

	Saturated Model	Estimated Model
SRMR	0.048	0.065
d_ULS	0.645	1.158
d_G	0.818	0.958
Chi-Square	415.655	473.765
NFI	0.844	0.823

**Table 7 insects-14-00547-t007:** R Square.

	R Square	R Square Adjusted
Individual beliefs	0.850	0.847
Intention	0.736	0.731
Perception	0.851	0.846
Social beliefs	0.390	0.385

Legend: Please see information in Table 2.

**Table 8 insects-14-00547-t008:** Collinearity statistics VIF.

Variable	VIF	Variable	VIF	Variable	VIF
Fam_condiments	3.512	Int_appearance	3.925	Prep_processed	1.650
Fam_daily	3.742	Int_familiarity	1.752	Prep_salty	1.663
Fam_experiment	1.902	Int_prepared	3.413	SB_friends&fam	1.768
IB_knowledge	4.310	Perc_familiarity	2.103	SB_soc_acceptability	2.196
IB_preparation	5	Perc_info	2.746	SB_try	2.241
IB_see	2.034	Perc_visual	2.642	Visual_presentation	2.198
Inf_environment	4.127	Prep_cooked	1.895	Visual_see	2.198
Inf_health	4.127	Prep_ingredients	2.170		

Legend: Please see information in Table 2.

## Data Availability

The data is available at request.

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
