# Peer review of "Nudging Consumer Behavior with Social Marketing in Portugal: Can Perception Have an Influence over Trying Insect-Based Food?"

_insects, 2023, doi:10.3390/insects14060547_

Round 1

Reviewer 1 Report

The manuscript presented to me for review entitled: Nudging consumer behavior with Social Marketing in Portugal: Can perception have any influence on trying insect-based 3 food? is an important element of the discussion about insect consumption, especially in Western countries.

Below are some suggestions to improve the manuscript.

[226] Why were only such variables used in the study? Whether the health aspect influences the decisions about consumption. In addition to behaviors and attitudes, acceptance is important. Acceptance and purchasing decisions carried out in the name of higher goals, ideas. consumers see it as a moral duty to the earth. I think that, this is crucial in the future strategy.

I realize that it is impossible to change the entire methodology, but it should be noted in the discussion.

[277] More and more often subjects have to choose between female, male and other. This is slowly becoming a standard, even if none of the respondents indicated such an answer.

[283] What was the final number of respondents.

The discussion is not exhaustive. There is no answer whether effective social marketing was used somewhere, for example in Europe. If not, what should be the main aims of such a strategy. In my opinion, the most important goal of this research is lost at the end.

[494] Promoting satiety is a very important and underestimated element.

Skotnicka, M., Mazurek, A., Karwowska, K., & Folwarski, M. (2022). Satiety of Edible Insect-Based Food Products as a Component of Body Weight Control. Nutrients 2022, Vol. 14, Page 2147, 14(10), 2147. https://doi.org/10.3390/NU14102147

Do these studies have application elements that can be used in the future. There have been many publications on the benefits and risks of eating edible insects. Unfortunately, there are few ideas and strategies on how to encourage society to consume. I missed that in this manuscript. Maybe this publication will be helpful.         

Mishyna, M., Chen, J., & Benjamin, O. (2020). Sensory attributes of edible insects and insect-based foods – Future outlooks for enhancing consumer appeal. Trends in Food Science & Technology, 95, 141–148. https://doi.org/10.1016/J.TIFS.2019.11.016

In my opinion, the conclusions are too long and [541-549] please move to the discussion section.

Author Response

Esteemed reviewer,

Thank you very much for your time and contributions. We really appreciate it.

We refined the text and amended it according to your suggestions.

Nevertheless, there are some contributions difficult to change in the manuscript. So, we will try to explain these cases. 

Thank you in advance for your understanding.

Please also see the attachment.

Reviewer 2 Report

This is a fascinating manuscript on a very interesting topic, in my opinion, but I believe several things should be modified.

1.  What is the clear objective of this work?  You stated it twice in sections 1 - 4 (Lines 73 - 87, Lines 235 - 236), it is confusing to the reader. 

2.  Sections #1 - 4 are excessive.  It is 5.5 pages TOO long!  This needs to be condensed and much of the information is repeated in the discussion where appropriate.  Suggestion - Get to your point, justify with some literature background the support for the experimental design/objective of the project, and utilize much of this information in the discussion. 

3.  The Materials and Methods are significantly lacking in detail and is only two pages long, compared to the four-part, five-page introduction.

4.  Since you are testing with humans, where is the information about ethics review and protocol approval?  There should have been an evaluation and the approval should be clearly cited in the paper.

5.  You had an online survey. What software was used to host the survey?  Based on your text, it wasn't Smart PLS software.  The survey ran from October 22 to March 23, but where was it hosted, and how did you get the subjects to take the survey?

6. The Smart PLS software analyzed the data, but how?  What were the statistical parameters at the foundation of the analysis? 

7.  There is bootstrapping done on the data, but the description of the bootstrapping is not listed in the M&M.

8.  Based on #3 - 7 above, I don't know how this work was completed and analyzed, which makes understanding the Results and Discussion sections incredibly difficult.

9.  Figure 1 (Source:  Own elaboration).  Should this be in the introduction section or the discussion?  

10 .  Figure 2.  This adds zero value to the manuscript.  It is a great communication with the research team, but not for the manuscript.

11.  You used Figure 3 and Table 3 to describe the demographics of the questionnaire participants.  NOWHERE did you indicate how many individuals took the questionnaire or if you had to do any cleaning of the data before analysis?

12.  Here is an excellent example of the lack of attention to detail.  In Figure 3, the age of the questionnaire respondents (18 - >55) was broken down into five age categories.  In Table 3, the mean age of the subject was 3.10.  Seriously?

13. Figure 4 and Figure 5 descriptions are useless.  It is the title only of the figure, but what am I looking at? What is the text in the yellow, blue, and green boxes?  What are the vectors and what do the numbers on those vectors mean?  There is no "breadcrumb trail" in the figure descriptions to help me, the reader, understand how to evaluate this information.

14.  Given the issues with Figures 4 & 5 and the lack of description in the Materials and Methods, I am unsure how to interpret or why the information in Tables 4 - 9 is important.  Again, their table descriptions are useless.

15.  I have no idea how a survey = Social Marketing.  You talk all about Social Marketing, but your tool was a survey.  This makes no sense to me and is misleading in this manuscript.  A survey is about gathering information; social marketing is about changing behavior.  The questionnaire you developed helps individuals interested in social marketing understand the triggers for each aspect of the behavioral model.  It does not do any social marketing.

16.  Also, you are not addressing any segmentation in the population regarding their perception of insect consumption or their willingness to consume insect based foods.  See the following:

Ho, I., Peterson, A., Madden, J., Wai, K., Lesniauskas, R., Garza, J.,  Gere, A.,  Amin, S., and Lammert, A.  2022.  The Crick-Eatery: A Novel Approach to Evaluate Cricket (Acheta domesticus) Powder Replacement in Food Products through Product Eating Experience and Emotional Response.  Foods.  DOI:  10.3390/foods11244115.

Rovai, D., Amin, S., Lesniauskas, R., Wilke, K., Garza, J., and Lammert, A.  2022.  Are early adopters willing to accept frozen, ready-to-cook mealworms as a food source?  J. Sensory Studies DOI: 10.1111/joss.12774

This manuscript needs to be reviewed for appropriate use of English.  Several grammatical/writing style errors must be fixed to improve the quality of the paper.  Here are some examples of line numbers that could use some assistance:  25, 30-31, 45-47,90, 103-105, 115, 150-152, 208,475, 519-520, 536.

Author Response

Esteemed reviewer,

Thank you very much for your time and contributions. We appreciate all your comments, requests and suggestions.

We have amended the manuscript as much as we could, so as to accommodate all these requests. We offered due explanations of all these changes in the message below. We also provided reasons in those cases where, unfortunately, a few requests could not be operated in the manuscript.

We are grateful in advance for your understanding.

Round 2

Reviewer 2 Report

1.  This paper still has a serious, ethical flaw.  There is no citation of the Internal Review Board Name and Number approving the survey instruments and protocol for this research.  If it is not cited, it violates the Declaration of Helsinki and should not be published.  I understand that the harm to humans is minimal, and no living data or biological samples were taken, but a third party must evaluate survey instruments and protocols.

Here are my comments based on the numbered of issues from my last review. The lack of attention to detail that I had a concern with before remains:

1. Corrected, thank you.

2.  Sections 1 - 4 Length:  Sections 1 - 4 are STILL too long, shorten.

3.  Short Materials & Methods:  a) Paragraph in lines 256 - 263 the use of the word aim, in different forms, 3 times.  Should be changed (b) You indicated that you validated answers, but did not say how.  How many people took the questionnaire, how did you validate it, and ultimately get to 110 answers?

4.  This is clearly addressed above.  You MUST have the IRB Name and Number for the approval of this work from an ethical perspective.  The language you put into the response to my comment is appropriate for the survey but is NOT sufficient without the IRB name and number listed in the manuscript.

5. Survey Software - (a) Thank you for adding the Google Forms information. (b) I looked at all three manuscripts you listed and the first two manuscripts do NOT have an IRB Name and Number listed in the manuscript (the third one appears to be a review paper).  Thus, those two papers are in ethics violation.  If your institutional IRB does not consider surveys as needing ethical review, you must provide a statement and URL from your IRB indicating this. (c) Do you have evidence that all Portuguese higher education schools have students over the age 18?  (d) Great you indicated that you validated the responses and are getting a fourth paper from the same data, why not reference how those results were validated from those other papers in this paper?

6.  Smart PLS - References 52-66 list how this method was used.  Great, but do you expect the readers of this manuscript to go back and read all of those articles to understand what you did?  You missed my point, you need to add language on how it was used in YOUR manuscript with details to YOUR manuscript while referencing those papers.

7.  Bootstrapping - (a) Great explanation, put it in the manuscript. (b) The figure you show in this justification is a HUGE problem.  It is a software printout, that is descriptive and I have no issue with that, but my issue is that the acronyms listed in the figure are not listed in the figure legend.  The reader has to play "search party" to figure out what these acronyms are in Figures 3 & 4.  List them in the legend as a courtesy to the reader.

8.  Understanding the Results & Discussion - See my response to #6 above.  If you don't add language for the readers to understand how you did the work, they are going to have trouble understanding the results and discussion, even with knowledge of this method.

9.  Figure 1 - Great justification.

10.  Old Figure 2 - Thanks.

11.  Figure 3/Table 3 - See my response to 3a.  "We validated ...."  How you may validate responses may be different than how your readers validate responses.  It requires an explanation by the authors please describe it.  

12.  Lack of attention to detail Figure 3/Table 3 - (a) You completely missed my point.  In Table 3, you indicate that the average age of your subjects was 3.1 (3.1 on a 5-point scale).  But in your response to me, you said "In this case, the most prevalent population was 46-55 years old."   So you are expecting the reader to know this?  Meaning that the reader needs to flip back through the manuscript, look at the age pie chart Figure 2, and determine that the third age classification is 46 -55.  Isn't the third age classification 36-45?  You did the descriptive statistics on the 5-point anchors, but in the table, it is not indicated as such, which leads the reader to believe it is age 3.  Again, you are expecting your readers to go back and forth between two types of representation of the same results to understand what you are doing. (b) In your response to me, there is no under-18 category that could be used for disqualification for survey participation.  This further supports why I am questioning the ethics evaluation of this work.  In our IRB evaluations, we are required to add this even though it is not likely to happen, but gives us the ability to delete the survey data if it is indicated.

13.  FIgure 4 & 5 descriptors - Not addressed.  See 7b above for further details.  Fam-_condiments, SB_friends&Fam ... Restate in the legend or in a stand-alone table to these acronyms, indicating that the language before the underscore can be found in XYZ location and the language after the underscore can be found in the ABC location. 

14.  Tables 4 - 9 - Again, your lack of attention to detail in deciphering the acronyms/codes used makes it incredibly hard to follow what you did.  See my response to 13 above.

15.  Survey=Social Marketing - Where did you add this clarifying language?  Is this the new text in the discussion?  I cannot assess where you did this and if it helped.

16.  Ho, et al - This reference was added in support of the segmentation/difference linkages that you found in your work, you misunderstood the spirit by which this reference was suggested.  It had nothing to do with the control and test group, but everything to do with supporting your findings.

Additional issues with this manuscript:

1. The section numbering is listed at 1, 2, 3, 4, 2, 3, 4, 5, is this acceptable?

2. There is no Data Availabity Statement provided.

There are a few small errors that could be corrected.

Author Response

Esteemed reviewer,

Thank you very much for your time and contributions. We appreciate all your comments, requests and suggestions.

We have amended the manuscript as much as we could, so as to accommodate all these requests, including a full revision of the English (language and grammar). We offered due explanations of all the changes related to the scientifical content in the message below. We also provided reasons in those cases where, unfortunately, a few requests could not be operated in the manuscript.

We are grateful in advance for your understanding.

________

Answers to Reviewer 2:

  1. This paper still has a serious, ethical flaw.  There is no citation of the Internal Review Board Name and Number approving the survey instruments and protocol for this research.  If it is not cited, it violates the Declaration of Helsinki and should not be published. I understand that the harm to humans is minimal, and no living data or biological samples were taken, but a third party must evaluate survey instruments and protocols.

Here are my comments based on the numbered of issues from my last review. The lack of attention to detail that I had a concern with before remains:

  1. Corrected, thank you.
  2. Sections 1 - 4 Length:  Sections 1 - 4 are STILLtoo long, shorten.

We rearranged this section and added supplementary information in another section. We hope in this moment the articles are balanced.

  1. Short Materials & Methods:  a) Paragraph in lines 256 - 263 the use of the word aim, in different forms, 3 times.  Should be changed (b) You indicated that you validated answers, but did not say how.  How many people took the questionnaire, how did you validate it, and ultimately get to 110 answers?

Thank you very much for your valuable recommendations. And thank you for having so much patience with us. We haven’t had the opportunity to meet such a professional reviewer by now, so we try to face this challenging situation.

We added supplementary information:

"We created a survey on this topic with multiple-choice questions, and the responses were analyzed. Although these results are relevant, it does not allow for a generalization of the results. For questions requiring multiple answers, we used the Likert scale. To apply the questionnaire, and taking into consideration our previous work, we considered the higher education students [53] at the Instituto Politécnico de Bragança (IPB) in Portugal. The chosen age was above eighteen years old. To assess the age of the sample we added a question about a range of age, which allowed us to select the following range of ages. This way we were able to control the age of the sample i.e.,   only students of eighteen years old or older were enabled to fill in the questionnaire. It was based on experiences with other universities [13, 14, 53]. However, only one hundred eleven records (answers from HEI) were verified.

As a result, this is only preliminary research, and in order to undertake secondary research, a more in-depth investigation on a representative sample is required. We utilized SmartPLS Software [54] version 3.0 (University of South Alabama, Mobile, AL, USA) to analyze the data, performing variable association and inferential analyses. To analyze the opportunities, difficulties, and utilities, we created a confirmatory factor analysis (CFA) approach.

We evaluated the effects of each element, including Intention to consume insects, Social Beliefs, Individual Beliefs, Perception of food, Visual aspect of the food, Familiarity with a specific kind of food, Preparation of food, Information acquired. By categorizing survey items into four variables (Table 4) and specifying the direction of their relationships, CFA enabled us to impose constraints on the model. The primary drawback of CFA is that it obscures the influence’s direction. Even so, it can quantify the effects of each element and subfactor, as well as demonstrate whether the model provides a good fit for the data and is consistent. The influence’s direction was also validated by literature review as we may see in the figure 1 that was a result of multiple research studies on food theme. Each variable’s loading factors are determined by the model. This was the primary justification behind choosing the CFA over a predictive analysis like a regression model [55].

Based on several metrics that improve how well the model explains the variables and matches the proposed hypothesis, the software will estimate the model saturation. In this regard, we can list comparative indices like Akaike's information criterion (AIC) and the Bayesian information criterion (BIC), as well as absolute indices like the normed fit index (NFI) and relative indices like the statistic value of chi-square (which allows inferential statistics) or standardized root mean square residual (SRMR). Cronbach's alpha was used to analyze the relevance of the specified latent components, while composite reliability, rho_A, and average variance extracted (AVE) were used to assess the model's consistency. The variance inflation factor (VIF) criterion was used to evaluate the model's multicollinearity."

"The survey was evaluated by a panel of experts in psychology. They did two types of validations: a) Qualitative pretesting (they tested if the content is understood, and if is applied to persons who are not the target of the questionnaire, and b) Quantitative pretesting (they tested if any adjustments may occur for the persons who are the target of the questionnaire). The Construct validity was based on reporting to already validated questionnaires, but we also applied a statistical approach: Pearson correlation coefficient for variables measured on report/ proportional scale, or Spearman correlation coefficient for variables measured on nominal or ordinal scale r. We also evaluated the fidelity and internal consistency of the questionnaire, checking if the items of the questionnaire constituted contribute to the constitution of the significance of a questionnaire and if the questions of the survey "go together" and mirror the same characteristic. A questionnaire is consistent when the items of which it is composed correlate, each of them, with the additive result of all the items. In this regard, we calculate the Cronbach-alpha coefficient and inter-item correlation matrix. In our case, the Cronbach-alpha coefficient is greater than the threshold of 0.70, meaning that our approach is correct. The data analysis used was the structure equation model with Smart PLS software [54, 55]."

  1. This is clearly addressed above.  You MUSThave the IRB Name and Number for the approval of this work from an ethical perspective.  The language you put into the response to my comment is appropriate for the survey but is NOT sufficient without the IRB name and number listed in the manuscript.

Thank you for the advice. We get IRB for our previous research developed in Romania (No 86 date 26.03.2021), with a similar survey, as well as No 135, dated 15.05.2021:

"Justified on the fact that it was a questionnaire applied to humans, the ethics review and protocol approval had a previous ethics committee opinion, for previous research developed in Romania, No 86 date 26.03.2021 as well as No 135 date 15.05.2021."

As our institutions, Spiru Haret University and Instituto Politehnico de Bragança, are partners in this project we based our research on this IRB.

  1. Survey Software - (a) Thank you for adding the Google Forms information. (b) I looked at all three manuscripts you listed and the first two manuscripts do NOT have an IRB Name and Number listed in the manuscript (the third one appears to be a review paper).  Thus, those two papers are in ethics violation.  If your institutional IRB does not consider surveys as needing ethical review, you must provide a statement and URL from your IRB indicating this. (c) Do you have evidence that all Portuguese higher education schools have students over the age 18? (d) Great you indicated that you validated the responses and are getting a fourth paper from the same data, why not reference how those results were validated from those other papers in this paper?

Thank you very much for your recommendation. We improved the paper:

(b): We are very concerned about ethics. We don't put the number because we think the respondent wouldn't understand, but we have the protocol. And we add the number in the methodology:

"Justified on the fact that it was a questionnaire applied to humans, the ethics review and protocol approval had a previous ethics committee opinion, for previous research de-veloped in Romania, No 86 date 26.03.2021 as well as No 135 date 15.05.2021."

(c ): As we explain it:

"To assess the age of the sample we added a question about a range of age, which allowed us to select the following range of ages. This way we were able to control the age of the sample i.e.,   only students of eighteen years old or older were enabled to fill in the ques-tionnaire. It was based on experiences with other universities [13, 14, 53]. Our previous researches empowered us to use a very similar survey, improved with different nuances. For example from the article “Sensory Perception Nudge: Insect-Based Food Consumer Behavior” we extracted only the information regarding sensory perception. From the article “Insect-Based Food: A (Free) Choice” we extracted information regarding individual and social beliefs. Based on all these elements, we established the intention of consuming insects and tried to find adequate social marketing approaches to stimulate this intention, by nudging consumer beahviour."
  1. Smart PLS - References 52-66 list how this method was used.  Great, but do you expect the readers of this manuscript to go back and read all of those articles to understand what you did?  You missed my point, you need to add language on how it was used in YOUR manuscript with details to YOUR manuscript while referencing those papers.

Thank you very much for your recommendation. A very important one!!! We improved the methodology section.

  1. Bootstrapping - (a) Great explanation, put it in the manuscript. (b) The figure you show in this justification is a HUGE problem.  It is a software printout, that is descriptive and I have no issue with that, but my issue is that the acronyms listed in the figure are not listed in the figure legend.  The reader has to play "search party" to figure out what these acronyms are in Figures 3 & 4.  List them in the legend as a courtesy to the reader.

Thank you very much. We included the text into the article.

"Bootstrapping is a re-sampling technique used to estimate the sampling distribution of one statistic. It involves repeated sampling from the original dataset with replacement to create multiple bootstrap samples. By analyzing these bootstrap samples, researchers can make inferences about the population from which the original dataset was drawn. Bootstrapping is particularly useful when the underlying distribution is unknown or when traditional assumptions about sampling distributions are not met.

Every construct's Variance Inflation Factor (VIF) was used to evaluate the relevance of each variable. VIF is a measure of multicollinearity in regression analysis. It quantifies the extent to which the variance of the estimated regression coefficients is inflated due to high correlations among the predictor variables. VIF values greater than 1 indicate the presence of multicollinearity, with higher values indicating more severe multicollinearity. High VIF values can lead to unreliable and unstable regression coefficient estimates. There is no evidence of collinearity between the variables because the VIF is lower than the recognized maxima limit of (5). To assess the importance of the variables, the Variance Inflation Factor (VIF) of each construct was run using 1000 samples. With the aid of SmartPLs software, bootstrapping reliability of 95% was accomplished [63]."

We understand very well your concern regarding legend for all figures and tables. In this regard we added a short legend (11 rows) for figure 3 and 4. In fact the entire legend is contained in table 2, which is very long. Would you be so kind as to give us an advice on how to include it? We thought a solution would be to add the text “Legend: Please see supplementary information in Table2” for each table and figure.

"Legend: Intention -The intention precedes behavior [49], and it is a choice with commitment [50], Social Beliefs -Social pressure as well as the recommendation from friends or colleagues, family can influence the intention to try insect-based foods [13], Individual Beliefs - Beliefs strengthens the attitude, and attitude leads to intentions and behavior [51], Perception - Consumers' perception [13, 36] is influenced by sensory liking [13, 37], Familiarity with a specific kind of food - Familiarity can be understood as information about the risks versus benefits [39] as well as the preference [40] or foods that the individual usually eats [41] because consumers are not inclined to taste food if they are not familiarized [38], Preparation of food - The preparation was considered an important factor that increases the predisposition to eat insect-based food. Being cooked [13], salty [38] or sweet, cold or hot, as well as the preparation methods and ingredients [13] were considered the main factors of influence, Visual aspect - Consumers prefer not to see the insect [13, 14, 38]. The color and presentation are important factors to try insect-based food [13], Information - The individual, social and environmental benefits should be added [38]. Supplementary information in table 2."

  1. Understanding the Results & Discussion - See my response to #6 above.  If you don't add language for the readers to understand how you did the work, they are going to have trouble understanding the results and discussion, even with knowledge of this method.
  2. Figure 1 - Great justification.
  3. Old Figure 2 - Thanks.
  4. Figure 3/Table 3 - See my response to 3a.  "We validated ...."  How you may validate responses may be different than how your readers validate responses.  It requires an explanation by the authors please describe it.  

Thank you. We added this explanation.

"The survey was evaluated by a panel of experts in psychology. They did two types of validations: a) Qualitative pretesting (they tested if the content is understood, and if is applied to persons who are not the target of the questionnaire, and b) Quantitative pretesting (they tested if any adjustments may occur for the persons who are the target of the questionnaire). The Construct validity was based on reporting to already validated questionnaires, but we also applied a statistical approach: Pearson correlation coefficient for variables measured on report/ proportional scale, or Spearman correlation coefficient for variables measured on nominal or ordinal scale r. We also evaluated the fidelity and internal consistency of the questionnaire, checking if the items of the questionnaire constituted contribute to the constitution of the significance of a questionnaire and if the questions of the survey "go together" and mirror the same characteristic. A questionnaire is consistent when the items of which it is composed correlate, each of them, with the additive result of all the items. In this regard, we calculate the Cronbach-alpha coefficient and inter-item correlation matrix. In our case, the Cronbach-alpha coefficient is greater than the threshold of 0.70, meaning that our approach is correct."

  1. Lack of attention to detail Figure 3/Table 3 - (a) You completely missed my point.  In Table 3, you indicate that the average age of your subjects was 3.1 (3.1 on a 5-point scale).  But in your response to me, you said "In this case, the most prevalent population was 46-55 years old."   So you are expecting the reader to know this?  Meaning that the reader needs to flip back through the manuscript, look at the age pie chart Figure 2, and determine that the third age classification is 46 -55.  Isn't the third age classification 36-45?  You did the descriptive statistics on the 5-point anchors, but in the table, it is not indicated as such, which leads the reader to believe it is age 3.  Again, you are expecting your readers to go back and forth between two types of representation of the same results to understand what you are doing. (b) In your response to me, there is no under-18 category that could be used for disqualification for survey participation.  This further supports why I am questioning the ethics evaluation of this work.  In our IRB evaluations, we are required to add this even though it is not likely to happen, but gives us the ability to delete the survey data if it is indicated.

Thank you very much for your recommendation. Expecting to help readers to understand better the results, table 3 was eliminated.

  1. FIgure 4 & 5 descriptors - Not addressed.  See 7b above for further details.  Fam-_condiments, SB_friends&Fam ... Restate in the legend or in a stand-alone table to these acronyms, indicating that the language before the underscore can be found in XYZ location and the language after the underscore can be found in the ABC location. 

We added the text “Legend: Please see supplementary information in Table2”. We hope this is an appropriate solution.

  1. Tables 4 - 9 - Again, your lack of attention to detail in deciphering the acronyms/codes used makes it incredibly hard to follow what you did.  See my response to 13 above.

We added the text “Legend: Please see supplementary information in Table2”. We hope this is an appropiate solution.

  1. Survey=Social Marketing - Where did you add this clarifying language?  Is this the new text in the discussion?  I cannot assess where you did this and if it helped.

Your concern is very relevant. Thank you very much for your recommendation.

As specialists, we consider more efficient aiming to nudge a specific consumer behavior on long term campaigns. Thus, social marketing is our supplementary perspective.

Like other surveys, our questionnaire referred to consumer behavior, which is understanding in the social marketing field. Aiming to clarify it, we added supplementary words such as “approach” or “campaigns” after social marketing.

"The key principles of social marketing include behavior changes and understanding the target audience. Aiming to achieve these objectives, social marketing campaigns employs a mix of marketing strategies that involves creating compelling messages, identifying appropriate channels for communication, and ensuring accessibility and affordability of desired behaviors. In addition, social marketing approach involves research and respective evaluation.

This article sets the focus of understanding the target audience to promote behavioral change goals. The main pillar that might be used in social marketing can be Nudging Consumer Behavior, a concept derived from behavioral economics that refers to subtly influencing people’s behavior without removing their freedom of choice. It involves persuasive techniques to guide individuals towards making choices. Nudging takes advantage of cognitive biases and heuristics that affect decision-making processes. In practice, nudging can be applied in various domains, such as encouraging sustainable behaviors (e.g., using energy-efficient appliances), promoting healthier lifestyles (e.g., encouraging exercise or healthy eating), and fostering financial well-being (e.g., promoting savings or responsible spending).

While answering the survey, the respondents were challenged to realize and understand the phenomena of solving feeding problems by using some insect proteins. The exposure might be a first step in social marketing campaigns with the aim of nudging their behavior.

In this case, perception plays a crucial role in consumers’ behavior and marketing. Consumers’ perceptions of products, brands, and marketing messages significantly influence their attitudes, preferences, and purchase decisions. Marketers aim to shape consumers’ perceptions through various strategies such as branding, advertising, packaging, and product positioning. Additionally, social marketing campaigns often rely on influencing individuals’ perceptions to promote behavior change because they seek to shape attitudes, beliefs, and social norms surrounding specific issues to encourage desired behaviors and discourage negative attitudes. Social marketing campaigns utilizes perception in different ways such as.

  1. Framing: Social marketing campaigns carefully frame the issue or behavior they are addressing to shape how it is perceived by the target audience. By highlighting certain aspects or emphasizing specific consequences, they aim to create a perception that motivates behavior change [69].
  2. Social Norms: Social marketing campaigns often leverages the power of social norms to influence perceptions. By highlighting the prevalence of certain behaviors within a social group or community, campaigns seek to create a perception of what is considered normal or desirable, thereby encouraging individuals to align their behaviors accordingly.
  3. Messaging and Communication: Effective social marketing campaigns employ persuasive messaging and communication strategies to shape perceptions. By using compelling narratives, emotional appeals, and relatable storytelling, they aim to create a perception that resonates with the target audience and motivates them to take actions.
  4. Social Proof: Social marketing campaigns often showcase positive testimonials, success stories, or endorsements from influential individuals or groups to influence per-ception. By providing evidence of others who have adopted the desired behaviors and ex-perienced positive outcomes, campaigns seek to establish social proof and encourage per-ception change."

/difference linkages that you found in your work, you misunderstood the spirit by which this reference was suggested.  It had nothing to do with the control and test group, but everything to do with supporting your findings.

We did miss the point. In this regard we added supplementary information regarding the intention of consuming insects.

"2.1. Behavior influence over consumption of insect-based food

In Thailand, it is common to eat insects. Because of their great nutritional value and affordable manufacture, entomophagy is considered as an alternative source of protein, but in many civilizations, consumer acceptability of insect-based cuisine remains a significant obstacle. People's propensity to consume insect-based dishes is influenced by social beliefs. Thus, these foods are rather consumed in social contexts (cafe, bar, pub, food festival), alone, and with social partners (friend, family, acquaintance, lover) [25, 26].

Entomophagy is influenced by perception factors such as visual aspects preparation and information (using names rather than images), and association with positive arousing emotions (joy, excitement, romance, tranquility. The adoption of innovative insect-based goods might be increased by including messaging about the advantages of eating insects during testing [27]. Consumer acceptability of insect-based foods may be raised by their perception [28].

The attitude, subjective norm, and perceived behavioral control of the subjects all contributed significantly (80%) to their desires to eat insect-based meals. Even with the goal to consume food derived from insects, culinary neophobia appeared adversely associated [27, 29, 30]. According to the theory of planned behavior (TPB), attitudes (A) are characterized both as favorable and unfavorable assessments of the behavioral intention. According to Ajzen (2005) [31] and Quine et al. (2000) [32], attitudes are driven by a person's views, individual's beliefs regarding the effects of own conduct and how strongly they hold those ideas and beliefs. The term "subjective norm" (SN) refers to how social conditioning affects the desired behavior. Individuals frequently assume that others close to them approve or disapprove of their planned behavior. People are more under social conditioning to alter their behavior if they think that those around them consider their intended behavior suitable.

The concept of perceived behavioral control (PBC) refers to people's perceptions (individual beliefs) of their capacity to carry out the desired behavior founded on past encounters or hearsay information from sources including friends, family, coworkers, and social networks, which is often greater the more people have access to resources and opportunities and the less they see barriers to changing their behavior [31]. Social marketing professionals engage in a deliberate strategic procedure designed to introduce one tiny community to innovative cuisine beforehand if readiness to ingest insect-based meals relies on embracing mnemonic shortcuts tied to cultural preconceptions. This ought to be carried out prior to spreading the novel food cuisine using educational programs to increase consumer awareness of consuming insect-based foods [33].

The future growth of entomophagy, as predicted by the respondents, as well as their intentions to try insects as food, the willingness to repeat an experience, the frequency of intake and the quantities ingested is real and it reveals different studies published so far. The near-universal knowledge (96%) of entomophagy, widespread prior experience of eating insects (67%) and near-universal desire to eat insects again among those who have already done so (90%) all indicate that the Western Europe may be a region where entomophagy has a very promising future. This is especially true for young men who care about the environment; however, product adoption presents difficulties due to issues with appearance, culture, and texture, requiring customized marketing strategies [34]. Penedo et al. [35] stated in a recent study that most participants in Switzerland indicated curiosity as the most likely justification for eating insect-based meals. Disgust was the most frequent pretext for avoiding such articles [35]."

Additional issues with this manuscript:

  1. The section numbering is listed at 1, 2, 3, 4, 2, 3, 4, 5, is this acceptable?

We did miss the point. In this regard, we review the numbers.

  1. There is no Data Availabity Statement provided.

We can provide any data if required. In this moment we can upload in the system the output of SmartPLS: FoodComplexFormative.xlsx

Round 3

Reviewer 2 Report

1.  Thank you for the numbering corrections.

2.  The added text throughout the entire document made the manuscript much easier to understand, most especially the attention to detail in the developing the instrument.

3.  Figure 3 - Remove the term "Legend" in each and add the language in lines 393 - 401 after line 404 and INCLUDED in the figure text.

4.  Figure 4 - Remove the term "Legend" in each and add the language in lines 529 - 537 after line 540 and INCLUDED in the figure text.

Lines 255 - 292 - This added text uses a lot of the term "we."  This should be changed to the third person.  I am fine with the content, my only concern is the language here.

Author Response

Esteemed reviewer and editors,

Thank you very much for providing us with the feedback on the submitted manuscript version. We are addressing each request in the message below. All changes operated in the manuscript following this review are also viewable in the track changes version.

_________

Reviewer 2:

Comments and Suggestions for Authors

  1. Thank you for the numbering corrections.

Re: The authors are grateful and thankful for the reviewers’ contributions.

  1. The added text throughout the entire document made the manuscript much easier to understand, most especially the attention to detail in the developing the instrument.

Re: The authors are grateful and thankful for all contributions that the reviewers have brought in improving this manuscript.

  1. Figure 3 - Remove the term "Legend" in each and add the language in lines 393 - 401 after line 404 and INCLUDED in the figure text.

Re: The term was removed a requested, and the explanation was included with the figure. The change can be viewed in the below print screen.

  1. Figure 4 - Remove the term "Legend" in each and add the language in lines 529 - 537 after line 540 and INCLUDED in the figure text.

 Re: The term was removed a requested, and the explanation was included with the figure. The change can be viewed in the print screen provided in the

above printscreen.

  1. Comments on the Quality of English Language

Lines 255 - 292 - This added text uses a lot of the term "we."  This should be changed to the third person.  I am fine with the content, my only concern is the language here.

Re: Thank you. In full agreement with the reviewer’s comments, the entire text of the manuscript was reviewed and changed accordingly.

All the requested changes are viewable in the track changes version of the manuscript (annexed).

________

Note to the editor with reference to revision of the IRBS part [i.e., “…please provide the name(s) of the ethics committee(s)/IRB(s) or other authorized body, besides the reason for exemption.”]:

Re: The Commission’s name is “Comisia de Etica – Universitatea Spiru Haret” (En. Ethics Commission – Spiru Haret University)